# Estimating Traffic Intensity Employing Passive Acoustic Radar and Enhanced Microwave Doppler Radar Sensor

**Andrzej Czyżewski**[ID]**, Józef Kotus**[ID] **and Grzegorz Szwoch** *[ID]

Telecommunication and Informatics, Multimedia Systems Department, Faculty of Electronics, Gdansk University of Technology, Narutowicza 11/12, 80-233 Gdańsk, Poland; ac@pg.edu.pl (A.C.); jozef.kotus@pg.edu.pl (J.K.)
* Correspondence: grzszwoc@pg.edu.pl

**Abstract:** Innovative road signs that can autonomously display the speed limit in cases where the traffic situation requires it are under development. The autonomous road sign contains many types of sensors, of which the subject of interest in this article is the Doppler sensor that we have improved and the constructed and calibrated acoustic probe. An algorithm for performing vehicle detection and tracking, as well as vehicle speed measurement, in a signal acquired with a continuous wave Doppler sensor, is discussed. A method is also experimentally presented and studied for counting vehicles and for determining their movement direction by means of acoustic vector sensor application. The assumptions of the method employing spatial distribution of sound intensity determined with the help of an integrated three-dimensional (3D) sound intensity probe are discussed. The enhanced Doppler radar and the developed sound intensity probe were used for the experiments that are described and analyzed in the paper.

**Keywords:** Doppler sensor; acoustic vector sensor; road traffic monitoring

## 1. Introduction

We develop innovative road signs that can autonomously determine and communicate (visually and over V2X, vehicle-to-everything radio messaging) the speed limit in cases where the traffic situation requires it in connection with the project that was carried out in our department. The project entitled "Intelligent Road Signs with V2X Interface for Adaptive Traffic Controlling" is carried out in response to the demand for improving road safety and traffic efficiency. The developed system of autonomous road signs will enable the prevention of the most common collisions on highways, resulting from the rapid stacking of vehicles that results most often from accidental heavy braking [1]. Figure 1 shows an example of a dangerous road situation, together with a system of autonomous road signs that display and wirelessly transmit (in the V2X standard) decreasing permissible speed as drivers approach the place with traffic obstruction.

The new design demands for solving various research and construction problems, such as effective and independent of weather conditions traffic monitoring based on simultaneous analysis of several types of data representation [2]. The engineering part of the project, as well as previous research results on this topic, as described in our earlier papers [3,4], were preceded by a series of experimental studies. Their results showed that measuring speed and traffic density causes a number of problems in practical conditions. For example, the use of visual analysis for this purpose encounters limitations that are associated with restrictions on the visibility of vehicles in both RGB and thermal cameras. The currently popular lidars also have some limitations and they are also relatively expensive. In this case, an estimation of the traffic exploits optical opacity of cars and laser beam reflections as the

physical principle of working is often accompanied by advanced data processing [5]. Setting the sensor perpendicular to the axis of the road makes it impossible to count vehicles in the case of occlusion (vehicles present on both lanes simultaneously), which causes many missed detections. Inductive loops or pneumatic cables for counting vehicles are used as a source of reliable data. However, their use requires installation in the road pavement, which is cumbersome and is only suitable for permanent installations, rather than for temporary installation of road signs in hazardous locations.

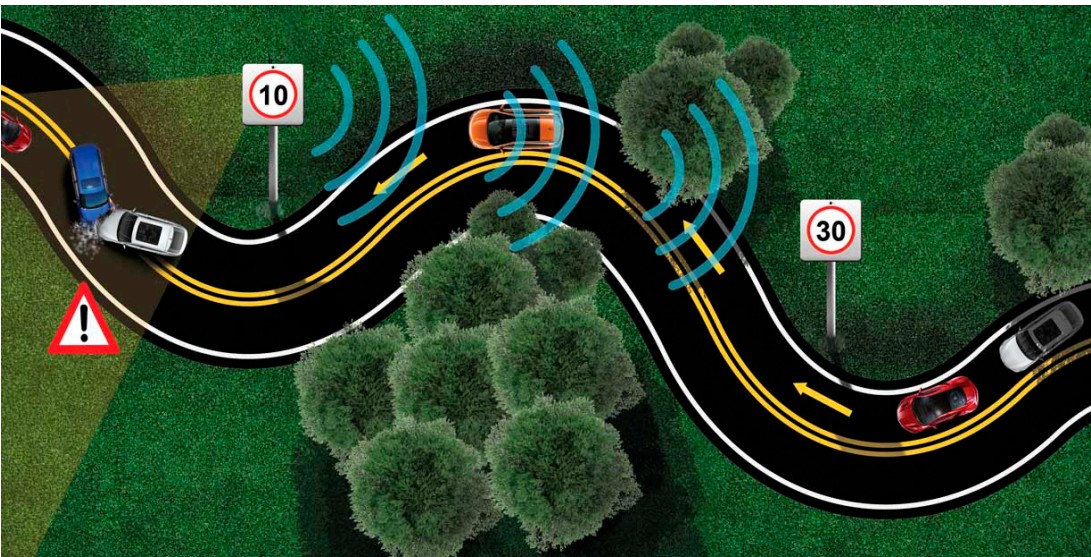

**Figure 1.** Illustration of the principle of limiting the speed when approaching a dangerous place on the road using autonomous road signs connected wirelessly to each other. Road signs communicate speed in a traditional way and also via radio channels using the V2X standard.

Another approach is based on recording anonymous Bluetooth MAC addresses of devices together with a timestamp as they pass by each detector, and then perform matching the addresses as vehicles pass through the next detector [6]. Specialized hardware was developed for related research (not covered in this paper), namely a radio module with software implementing the baseband controller and the firmware link manager layers. We constructed a practical vehicle counter in this way and then performed field tests showing that this technology makes a promising method of collecting real-time statistical traffic data and actual journey times from measurements on long distances, e.g., 1 km, which is not possible or difficult with other modalities [4]. However, counting vehicles that are based on the MAC addresses of Bluetooth devices can be unreliable, because these addresses are associated with both vehicles and mobile phones that are used by pedestrians. There can be several Bluetooth devices in one vehicle, e.g., audio and communication devices, as well as radio modules that are embedded in-vehicle diagnostic systems. Gupta et al. presented a different and interesting idea. They proposed the system for vehicles (i.e., bicycles, cars, trucks) counting by means of variations in the Wi-Fi signals strength [7]. Another approach that was related to the monitoring of traffic flow exploited the magnetic sensors to measure the vehicle's magnetic signature (VMS) evoked by moving vehicles [8].

Still, measurements of traffic intensity and vehicle speed while using Doppler microwave sensors find technical and economic justification. For example, a system that is based on high range resolution based on microwave radar sensor has been previously proposed by other authors to estimate the traffic flow rate and the flow rate of certain types of the vehicle [9,10]. However, microwave sensors are exposed to interference due to noise in the radio channels and reaching the receiver by parasitic reflections and microwave interferences. For this reason, we worked on this issue and presented an experimentally verified approach in this paper that allows for improving the results that were obtained while using a microwave radar.

Acoustic methods of road traffic estimation are, in practice, quite rarely used, although experimental research in this field is being conducted [11–13]. Our department has also been conducting such research for some time, which resulted in publications [14,15] and a recently defended doctoral dissertation [16].

A method is presented and experimentally studied for counting vehicles and determining their movement direction by means of the acoustic vector sensor application and the enhanced Doppler microwave sensor. The assumptions of the method employing spatial distribution of sound intensity determined with the help of an integrated three-dimensional (3D) intensity probe are discussed. The developed intensity probe was used for the experiments that brought the results discussed in the paper.

## 2. Materials and Methods

### 2.1. Vehicle Counting and Speed Measurement with Doppler Sensor

#### 2.1.1. Doppler Sensor

The sensor that was used in the presented research emits a continuous wave with constant frequency within the K band (24.125 GHz) and it provides a dual-channel (I/Q) signal with frequencies below ca. 8 kHz, being proportional to the object's velocity, according to the Doppler effect. The sensor is characterized by a wide horizontal beam, which allows for capturing a vehicle's movement within a sufficiently long road segment (at least 50 m). This is different from the majority of radar sensors used in practical measurement systems, which measure the vehicle speed within a narrow zone.

The sensor transmits an electromagnetic wave with a constant frequency $f_0$. The frequency $f_r$ of waves reflected from moving objects and received by the sensor differs from $f_0$, according to the Doppler effect [17,18]. An I/Q mixer produces a difference signal with frequency $f_d$, in two channels: in-phase (I) and quadrature (Q), which allows for the detection of the object's direction of movement (phase difference between I-Q channels is either 90° or −90°). The frequency $f_d$ is related to the object's velocity $v_r$ by an equation:

$$f_d = |f_r - f_0| = \frac{2}{\lambda} v_r = \frac{2f_0}{c} v_r = S v_r, \tag{1}$$

where $c$ is the speed of light. For a K-band sensor, $f_0$ = 24.125 GHz, and the scaling factor $S \approx 160.94$ ($v_r$ in m/s). $f_d \leq 8.94$ kHz for road vehicles moving with speed up to 200 km/h (55.5 m/s). Therefore, the difference signal fits in the audio band (it is indeed audible), so standard audio signal processing algorithms may be applied for vehicle detection and speed measurement.

For practical reasons, it is not possible to directly mount the sensor on the vehicle's path of movement, therefore the sensor is usually placed alongside the road (Figure 2). As a result, the sensor only measures the radial component $v_r$ of the velocity vector. As a vehicle moves through the detection zone of a sensor, $v_r$ decreases when the vehicle approaches the sensor, and then increases as it moves away. This is called a 'cosine effect' [19], as the actual velocity is multiplied by a cosine of the angle to the object (Figure 2). Additionally, the angle difference between the front and the rear of the vehicle becomes larger as the vehicle moves closer to the sensor, which results in $v_r$ values spanning a wider range. Figure 3 illustrates this, showing a spectrogram of a signal reflected from a road vehicle and recorded by a Doppler sensor. Signal frequency spans a range ($f_{\min}$, $f_{\max}$), given by:

$$f_{\min} = Sv \cos \alpha_{\max} = Sv \frac{r}{\sqrt{r^2 + y^2}} \tag{2}$$

$$f_{\max} = Sv \cos \alpha_{\min} = Sv \frac{r+d}{\sqrt{(r+d)^2 + y^2}} \tag{3}$$

where Figure 2 explains $r$, $d$, $y$, and $\alpha$.

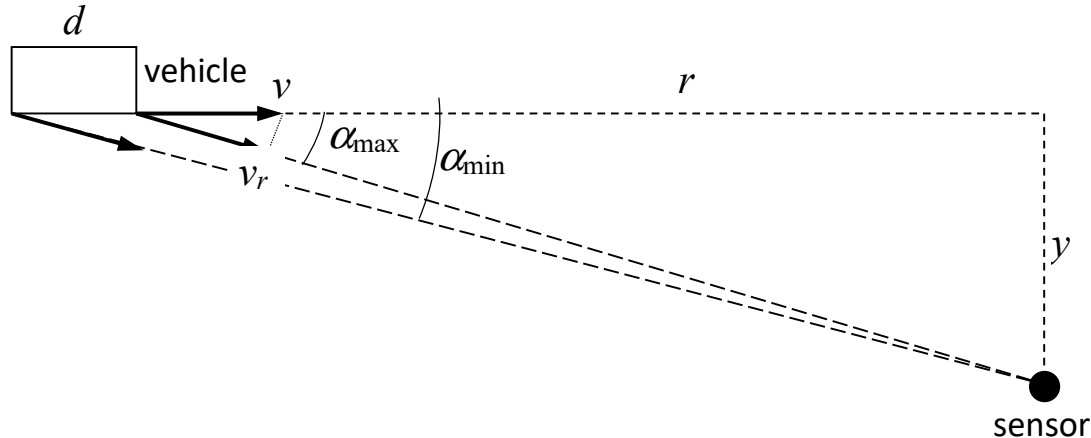

**Figure 2.** Measurement of the radial velocity of a vehicle with a Doppler sensor.

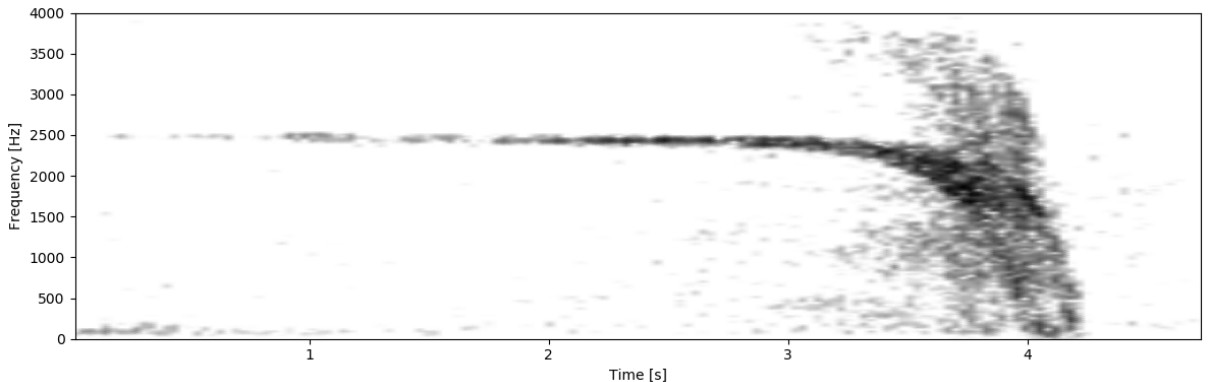

**Figure 3.** Spectrogram of a Doppler sensor signal recorded when a single vehicle was moving towards the sensor. The cosine effect is visible in the final phase (time 2.5–4.5 s).

Compensation of the cosine effect is problematic. Therefore, frequency is usually taken from the signal part captured at a large distance between the sensor and the object for speed measurement, so that the angle $\alpha$ is small (cos $\alpha$ close to 1), the difference between $\alpha_{min}$ and $\alpha_{max}$ is also small, and therefore the cosine effect is negligible. This approach was used in the proposed algorithm.

### 2.1.2. Algorithm for Processing of Doppler Sensor Signals

The task of the algorithm is to perform vehicle detection and tracking, as well as vehicle speed measurement, in a signal that was acquired with a continuous wave Doppler sensor. Figure 4 shows an overview of the processing algorithm. A dual-channel signal is received from an I/Q Doppler sensor. The first stage is the signal preprocessing, which suppresses noise and interference in the signal and then decomposes the signal into two components that represent opposite directions of movement. In the next stages, signal components that are reflected by moving vehicles are detected, and tracking of individual vehicles is performed. Finally, a velocity estimate is calculated from each identified vehicle track. The details of the algorithm are presented in the following Subsections.

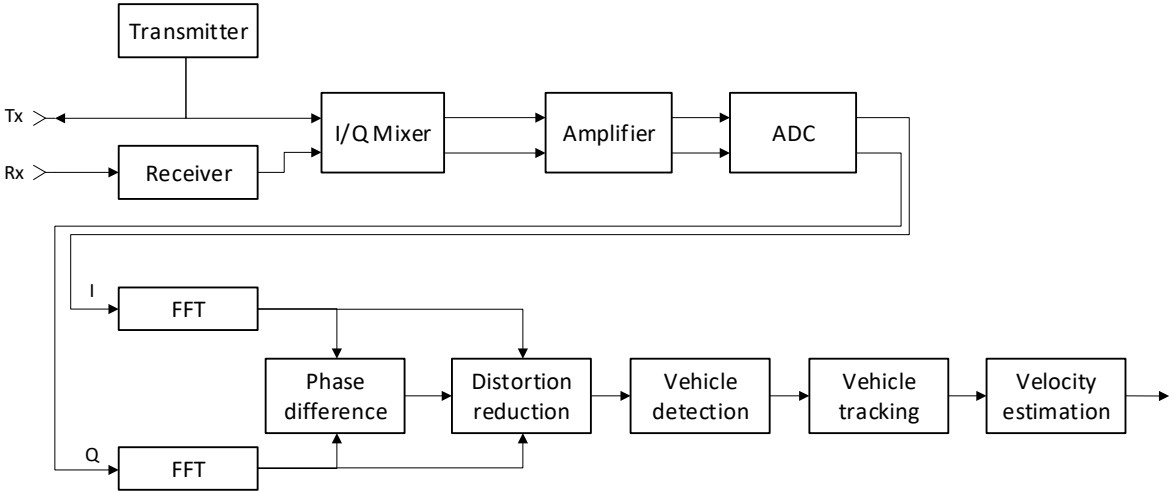

**Figure 4.** Block diagram of the Doppler sensor and the processing algorithm.

2.1.3. Suppression of Interference and Noise

The signals obtained from a Doppler sensor are often contaminated with distortions that make the detection and tracking processes more troublesome (Figure 5a). There are two main types of distortions that are observed in Doppler sensors. Noise is present in the signal as wide-band spectral components with random amplitude and phase, as a result of sensor imperfections, the nature of wave reflection, and environmental factors, such as wind. Electromagnetic interference (EMI) usually manifests as narrowband spectral components with a constant frequency. They may be induced by air (e.g., from nearby radio frequency transmitters, such as mobile network stations or airport radars) and by power lines. The amount of signal distortion depends on the sensor class, its positioning, and orientation, the quality of power supply, etc. As a result, a procedure for the suppression of distortions is necessary before the detection and tracking are performed.

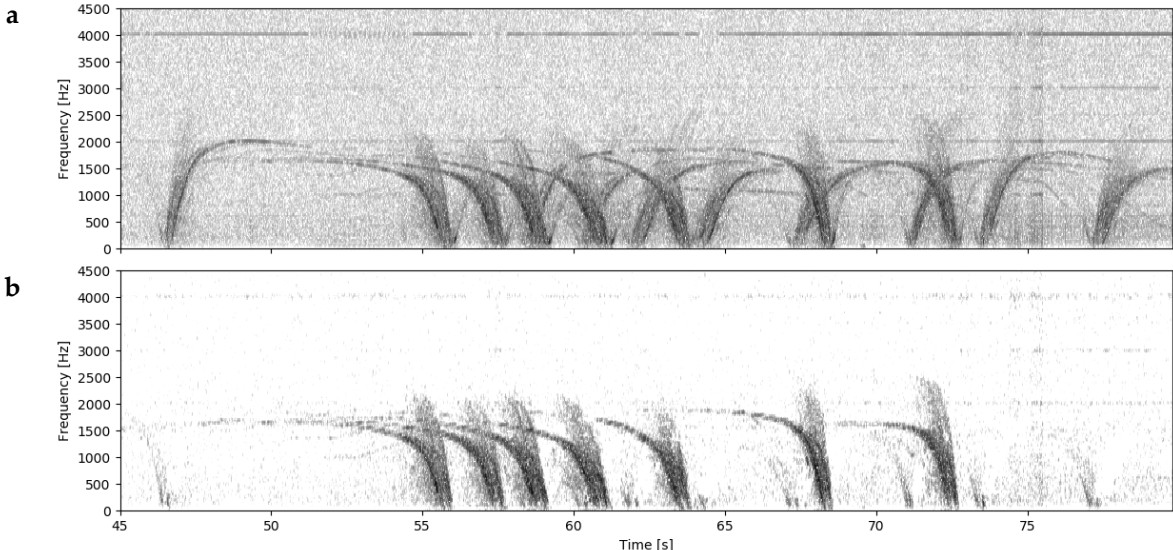

**Figure 5.** Spectrograms of road traffic signals: (**a**) signal recorded by the Doppler sensor, with electromagnetic interference (EMI) at multiplies of 1 kHz, and a wideband noise, (**b**) processed signal—suppression of noise, interference, and signals reflected from vehicles moving away from the sensor.

Noise reduction is usually performed by computing a noise profile and subtracting it from the signal. Such an approach requires the detection of signal parts containing only noise and constantly updating the profile. It is also not efficient for EMI removal. Therefore, a novel approach, which is

based on the phase relationship between I/Q channels, is proposed. An additional benefit of this algorithm is the separation of the opposite directions of movement [20]. The algorithm is based on the phase difference $\Delta\phi$ between the I/Q channels of the sensor signal:

$$\Delta\phi = \arg Q(\omega) - \arg I(\omega) \tag{4}$$

where $Q$ and $I$ are the spectra of the respective signal channels.

In theory, $\Delta\phi$ should be equal to $\pm 90°$ for signals that are reflected from moving objects. In practice, $\Delta\phi$ varies within a range, depending on the sensor class. The following observations were made from the analysis of phase in I/Q sensor signals [20]:

- signal components reflected from objects approaching the sensor or moving away from it have $\Delta\phi$ following a normal distribution with mean equal to $90°$ or $-90°$, respectively;
- for the noise, $\Delta\phi$ has a normal distribution with the mean value close to $0°$ and it might overlap the signal parts, depending on the sensor class; and,
- EMI is concentrated around $\Delta\phi = 0°$, as it influences both I/Q channels in an identical way.

Therefore, the algorithm for suppression of noise and EMI is based on the concept of a 'phase filter', as illustrated in Figure 6. The signal spectrum is multiplied by a weighting function $w$ given by:

$$w(\omega) = \frac{1}{1 + e^{-\gamma(u(\omega) - 0.5)}} \tag{5}$$

where $u$ is given by:

$$u(\omega) = 1 - \left| \max\left( \frac{\Delta\phi(\omega)}{90}, 0 \right) - 1 \right| \tag{6}$$

for objects moving towards the sensor ('oncoming'), and

$$u(\omega) = 1 - \left| \max\left( \frac{-\Delta\phi(\omega)}{90}, 0 \right) - 1 \right| \tag{7}$$

for objects moving away from the sensor ('outgoing'), where $\Delta\phi$ is expressed in degrees [20]. The $\gamma$ parameter controls the shape of the weighting function. As shown in Figure 6, part of the signal energy is lost if $u$ is used as the weighting function (the 'no $\gamma$' case), and the overlap with the noise distribution is larger than in the case of additional shaping of the function. In the experiments, the authors found that $\gamma = 20$ is optimal, providing a proper balance between signal preservation and the suppression of distortions.

Figure 5b shows the example results of the preprocessing while using the proposed algorithm. As can be seen, EMI that occurred at multiplies of 1 kHz (most prominent at 4 kHz) is almost completely removed and wideband noise is significantly suppressed. The remaining speckle-noise results from the partial overlapping of signal and noise distributions, and from 'reflected' signals, captured when a vehicle has already passed the sensor. Such remaining noise components are discarded with amplitude thresholding in the later stage of processing. As can be seen, two opposite directions of the movement were separated, so that they may be individually analyzed. The detection and tracking phases are now significantly easier to perform due to the removal of occlusion by objects that are moving in opposite directions.

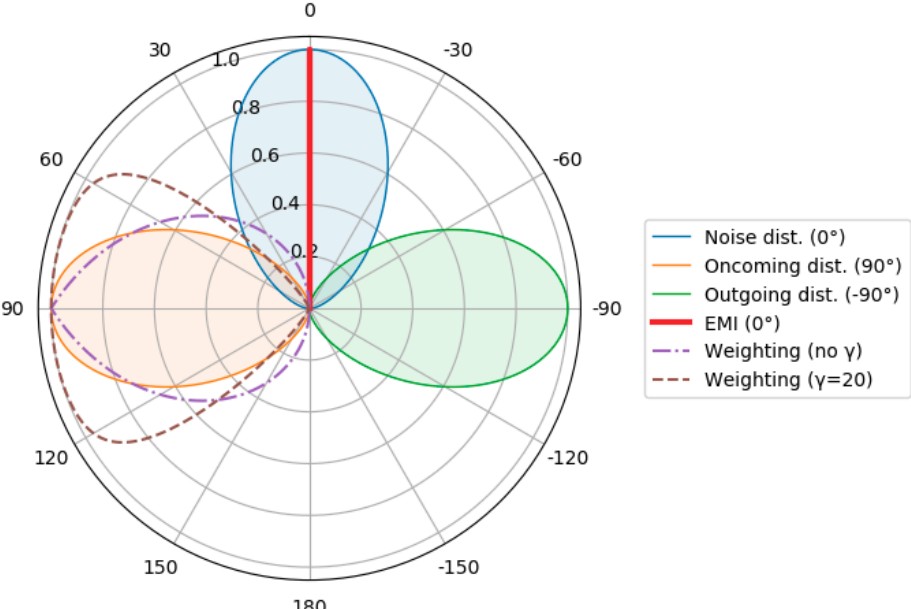

**Figure 6.** Polar plot illustrating the concept of phase filtering. Filled areas show distributions of signal and noise ($\sigma = 30°$), dashed lines show the shape of the weighting functions. The outer axis is scaled in the phase difference between I/Q channels (in degrees), the radial axis shows relative amplitude or gain.

### 2.1.4. Vehicle Detection, Tracking, and Velocity Estimation

The automatic measurement of the velocity of vehicles in road traffic, being performed by an unsupervised algorithm, requires performing three stages. First, the detection of spectral components that represent moving vehicles is performed in the preprocessed signal, in blocks of signal samples (short term analysis). In the second stage, the detection results have to be assigned to vehicles, so that changes in the signal frequency (caused by the object movement and the cosine effect) are tracked. The preprocessing algorithm that is presented in the previous Subsection, by decreasing the level of noise and interference, allows for easier vehicle detection, and eliminating occlusion from vehicles moving in the opposite direction allows for easier tracking. Occlusion from vehicles moving in the same direction still occurs and it is the main problem in the tracking. In the final stage of the analysis, the estimated velocity is extracted from each vehicle track. It should be noted that the velocity measurement with an automatic algorithm is much more problematic than where a human operator is able to relate the measurements to the observed vehicles.

The detection algorithm works on signal spectra that were computed in short windows (e.g., 2048 samples, 42.6 ms for 48 kHz sampling), after multiplication by the weighting function, as described earlier. The detection works by finding sequences of spectral bins with an amplitude above a threshold (that should be set according to the signal level and the remaining noise level). The threshold should be sufficiently low in order to detect weak signals when a vehicle is far from the sensor. Groups of spectral bins containing the reflected signal become larger when a vehicle approaches the sensor (Figure 3). In practice, gaps occur in such groups (due to weaker signal components at some frequencies), and such gaps result in the segmentation of signal parts. These fragments have to be merged in the detection phase. It is also inevitable that some strong noise components are incorrectly detected as signals.

The tracking algorithm merges the detection results from consecutive time windows. Each stored track is extrapolated to find the expected frequency in the current window. Linear interpolation is used in the initial phase and cubic interpolation in the later stage for objects approaching the sensor. Next, the detected groups of spectral bins are searched for the one closest to the expected value. If such a group is found, this group is appended to the track, its centroid frequency is computed, and the lowest and the highest frequency is stored in the track. The track is not updated when no matching group is found. Additionally, dead tracks are removed from the analysis, and finished tracks (i.e., with a

sufficient length, and with a minimum frequency below a threshold of ca. 500 Hz, below which the vehicle is very close to the sensor) are passed to the velocity estimation phase.

The velocity of a vehicle should be measured within an initial ('thin' and flat) section of a track. Therefore, a finished track is analyzed in short sections. Frequency computed in each window is converted to velocity, and then the mean and the standard deviation are computed for each section. The algorithm selects a section with the highest mean, provided that the standard deviation is below the threshold. The latter condition allows for the elimination of tracking errors (track following a group of noise components) from velocity estimation. The computed standard deviation is used as a representation of the 'quality' of the result. If the value is obtained from an initial, flat section of a track, and the standard deviation is low (<0.1 km/h). The value is obtained from the later track part (with decreasing frequency), which will be reflected by the higher standard deviation, if there is no flat segment, e.g., due to occlusion by another vehicle ahead. The results with a high standard deviation (>1.0 km/h) should be discarded.

Figure 7 shows an example of a successful analysis in a signal recorded during the experiments. Only vehicles moving on the lane closer to the sensor (the oncoming traffic) were analyzed. The results of the detection stage are shown as gray points. Some of the detected components are assigned to the tracks of individual vehicles, color dots mark the computed centroids of the detected spectral bin groups, thus forming continuous tracks. The obtained velocity estimates are also shown in the plot. All of the estimates had a standard deviation not exceeding 0.25 km/h.

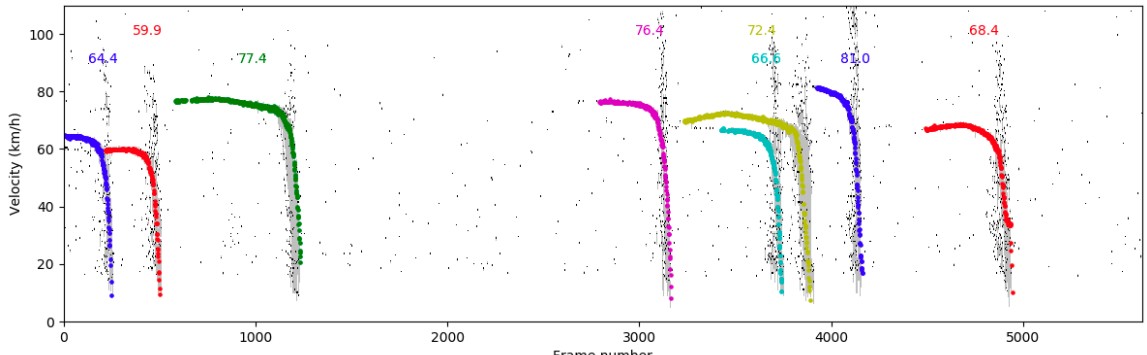

**Figure 7.** Example of vehicle detection and velocity measurement with the proposed algorithm. Gray points indicate the detection results, color dots mark the detection results assigned to the tracks of individual vehicles (distinguished by a color), values show the obtained velocity estimates (km/h).

## 2.2. Vehicle Counting and Speed Measurement with Acoustic Vector Sensor

### 2.2.1. Acoustic Vector Sensor

An intensity-based AVS (Acoustic Vector Sensor) described in this paper consists of three pairs of pressure sensors (microphones) positioned on three orthogonal axes, in equal distances from the center point. Each pair of microphones forms a *p-p* sensor, which is used for measuring the particle velocity in a given direction. The averaged pressure at the center point is computed as a mean of all six pressure signals. The intensity on each axis is proportional to the product of the particle velocity and the averaged pressure. Figure 8 shows the AVS used during experiments. Computation of intensity and angles are performed by the algorithm, as outlined in Figure 9. The developed algorithm for source localization and tracking has three main sections. The first part is related to the correction section. Correction of the pressure signals, as obtained by the microphones, should be applied for the proper determination of the sound intensity. This correction is realized in two steps. First, the frequency responses of all microphones are equalized, and the phase response in the microphone pair is also equalized (block denoted as *A&P.Corr* in Figure 9). Next, the particle velocity on each axis ($ux$, $uy$, $uz$) and the average acoustic pressure $p$ are calculated. Then the second step of the correction is performed

in which phase differences between the particle velocity and the average acoustic pressure signals are equalized (*P.Corr*). The authors' previous publication described the detailed description of the calibration and correction process [21].

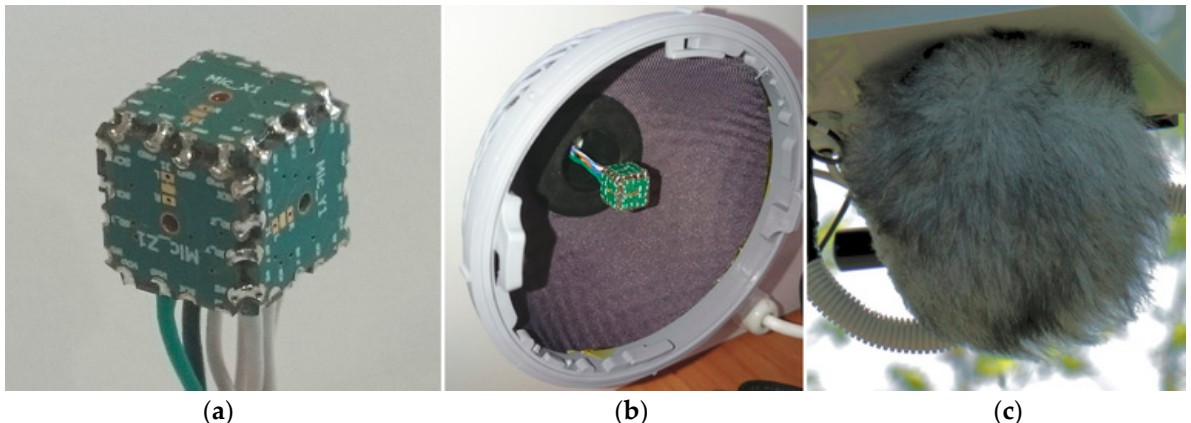

| (a) | (b) | (c) |

**Figure 8.** The acoustical vector sensor (AVS) designed by the authors (**a**), the AVS inside the windscreen (**b**), and the AVS during the measurements (**c**).

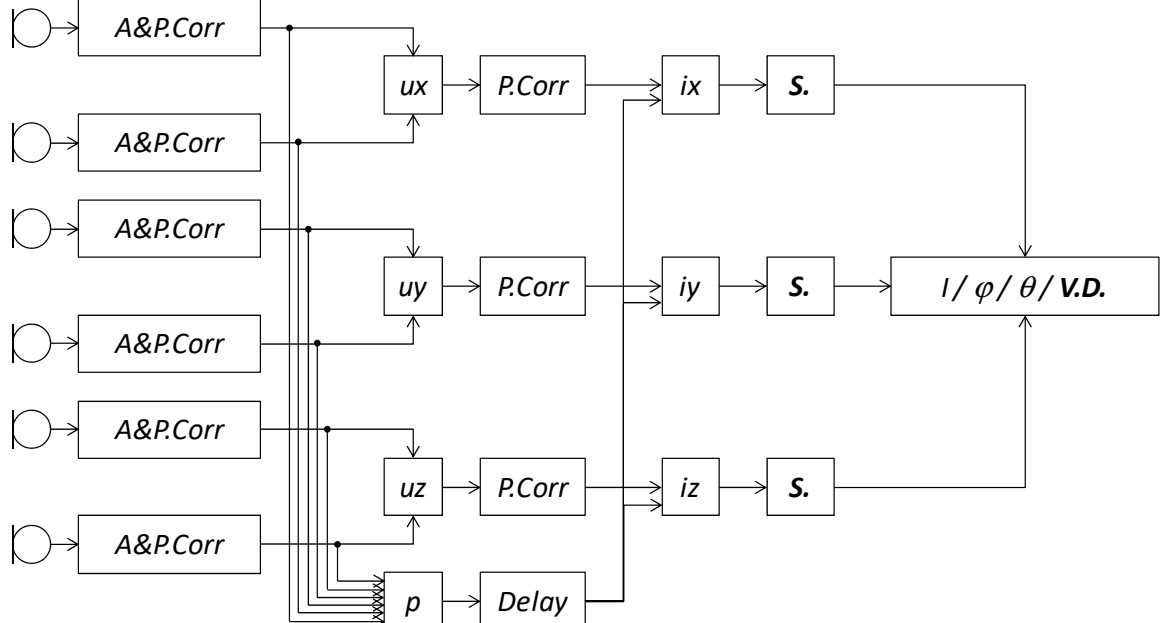

**Figure 9.** Block diagram of the developed algorithm for detection, localization, and tracking of moving sound sources.

After these steps, the sound intensity components (*ix*, *iy*, *iz*) can be determined by computing a product of each intensity component with the averaged pressure signal and by integrating the result [22, 23]. The second part of the algorithm contains smoothing blocks, labeled as *S*. Noise suppression procedure has to be applied to each intensity signal in order to make the vehicle detection possible. In the experiments described in this paper, a Savitzky–Golay filter was used in order to suppress noise and to obtain smoothed intensity functions [24]. The optimal values of the filter length (51) and the polynomial order (3) were experimentally found, as a good balance between the details and noise present in the processed signals. At the end of the second part, the three smoothed components $I_X$, $I_Y$, and $I_Z$ are known.

The last section, labeled as *V.D.*—vehicle detection block—uses the intensity components and values of azimuth and elevation angle for making a decision regarding the presence of sound source (a vehicle in the considered scenario).

### 2.2.2. Intensity Computation

A three-dimensional (3D) AVS applied during our research is able to measure the acoustic particle velocity (referred to as "velocity" further in the text) in three orthogonal directions, as well as pressure in the central point of the sensor. Sound intensity vectors in three orthogonal directions may be obtained based on the velocity and pressure measurement results.

The acoustic pressure has to be measured at six points located on three orthogonal axes, at identical distances $d$ from the origin. These points are denoted as $x1$, $x2$, $y1$, $y2$, $z1$, and $z2$, describing their location in the coordinate system, e.g., point $y2$ is located at $(0, d, 0)$ and $y1$ at $(0, -d, 0)$. Omnidirectional microphones of the same type are used to measure pressure $p_l(t)$ at six locations $l$. According to the Euler's formula [22], velocity vectors $\mathbf{u}_i(t)$ alongside axes X, Y, Z may be computed as:

$$
\begin{bmatrix} \mathbf{u}_x(t) \\ \mathbf{u}_y(t) \\ \mathbf{u}_z(t) \end{bmatrix} = \begin{bmatrix} a_x & 0 & 0 \\ 0 & a_y & 0 \\ 0 & 0 & a_z \end{bmatrix} \cdot \begin{bmatrix} p_{x2}(t) - p_{x1}(t) \\ p_{y2}(t) - p_{y1}(t) \\ p_{z2}(t) - p_{z1}(t) \end{bmatrix}
\tag{8}
$$

where $a_i$ are the scaling factors (determined during calibration procedure). The magnitude of the $\mathbf{u}_i(t)$ vector will be denoted as $u_i(t)$. Pressure $p(t)$ measured at the origin is averaged from two points at the given axis and it has to be equal on all three axes. In practice, the pressure is averaged from all six microphones (as in Equation (9)):

$$
p(t) = \frac{p_{x1}(t) + p_{x2}(t) + p_{y1}(t) + p_{y2}(t) + p_{z1}(t) + p_{z2}(t)}{6}
\tag{9}
$$

The sound intensity $I$ at a given axis can be then computed, as [22]:

$$
I = \frac{1}{T} \int_T p(t)u(t)\,\mathrm{d}t
\tag{10}
$$

where $T$ is the integration period.

If a single, omnidirectional sound source is put into the system at polar coordinates $(r, \phi, \theta)$, the angles of the sound received by the AVS may then be computed as:

$$
\phi = \arctan\left(\frac{I_y}{I_x}\right)
\tag{11}
$$

$$
\theta = \arctan\left(\frac{I_z}{\sqrt{I_x^2 + I_y^2}}\right)
\tag{12}
$$

where $I_x$, $I_y$, $I_z$ are the intensity signals measured along the axes of the coordinate system, being oriented as shown in Figure 10.

In general, the sound intensity is determined by means of the algorithm described above in the time domain while using broadband signals of acoustic pressure and particle velocity. For purposes that are considered in this article, it is important to perform sound intensity calculation in the frequency range related to the acoustic events produced by vehicles moving near the sensor. The frequency analysis of the background noise and pass-by vehicle sounds were performed to avoid the unwanted and disturbing sounds emitted by other sound sources during vehicle movement. Figure 11 shows the results of this analysis. The dotted line indicates the background noise. The solid line depicts

the spectrum of pass-by vehicle event. It can be noticed that for frequency greater than 6 kHz the background noise is close to the noise that is emitted by the vehicle. The grasshoppers generated this high background noise. Low-frequency noise can be produced by the wind and other sound sources placed far away from the measurement point. For this purpose, the sound intensity analysis was limited to the frequency range: 400 Hz–4 kHz. It was shown in Figure 11 while using two vertical dotted lines. In this way, the essential part of the acoustic energy emitted by the moving vehicle was taken into consideration during the calculation of sound intensity and direction of arrival.

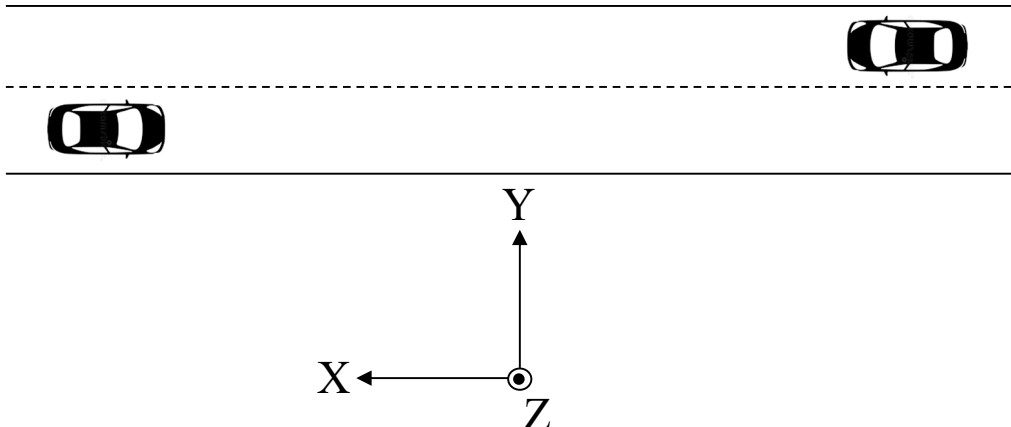

**Figure 10.** Orientation of the acoustic vector sensor relative to the road.

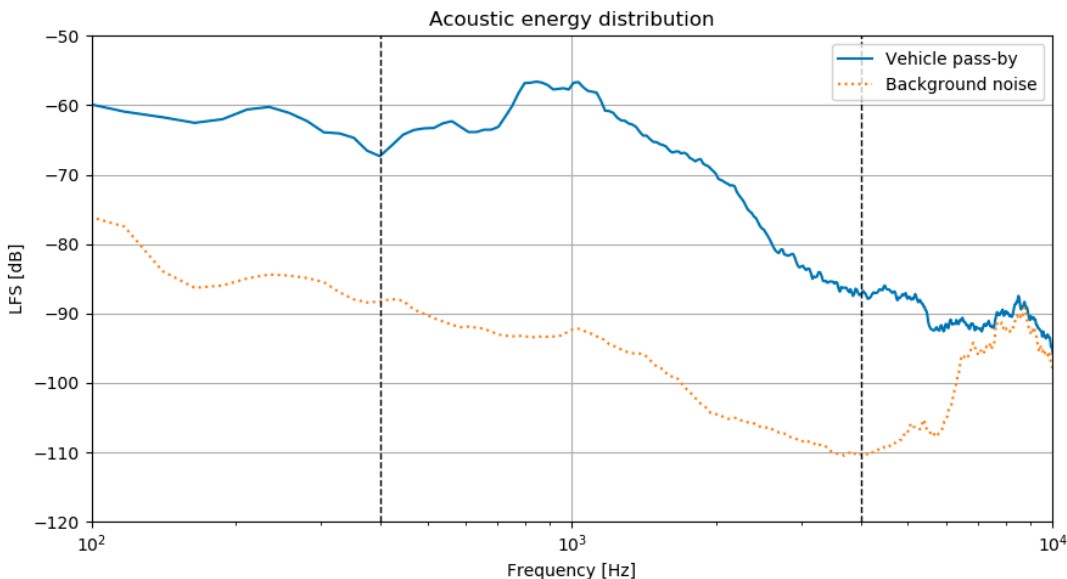

**Figure 11.** Acoustic energy distribution in the frequency domain for background noise and for the pass-by vehicle.

Figure 12 shows an example of the output of the algorithm described above for an acoustic event evoked by a single pass-by vehicle. The left chart depicted the sound intensity components. We can indicate the highest values that were obtained for the direction perpendicular to the road ($Iy$). Other components have relatively lover values. This observation will be an essential fact during the development of the vehicle detection module. The right chart includes the direction of arrival components, being expressed by azimuth and elevation angles. In Figure 13, an example of 120 s of sound intensity continuous analysis was shown. The acoustic events that are evoked by the vehicles are clearly visible. An event typical for a group of vehicles occurred around 80 s. The rapid changes of the azimuth angle can be noticed for this event. It is important to emphasize that no other acoustic

events than passing vehicles can be observed. It confirms that the frequency range of sound intensity calculation was correctly selected.

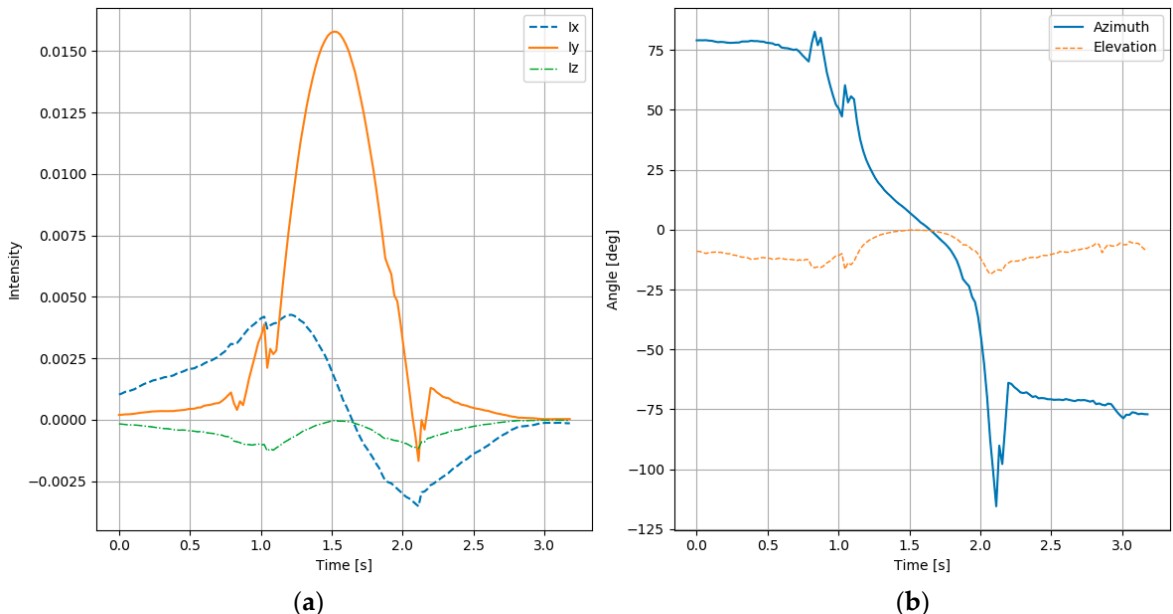

**Figure 12.** The results obtained from the AVS signal recorded for a single-vehicle: (**a**) intensity components, (**b**) azimuth and elevation.2.2.3. Vehicle Detection.

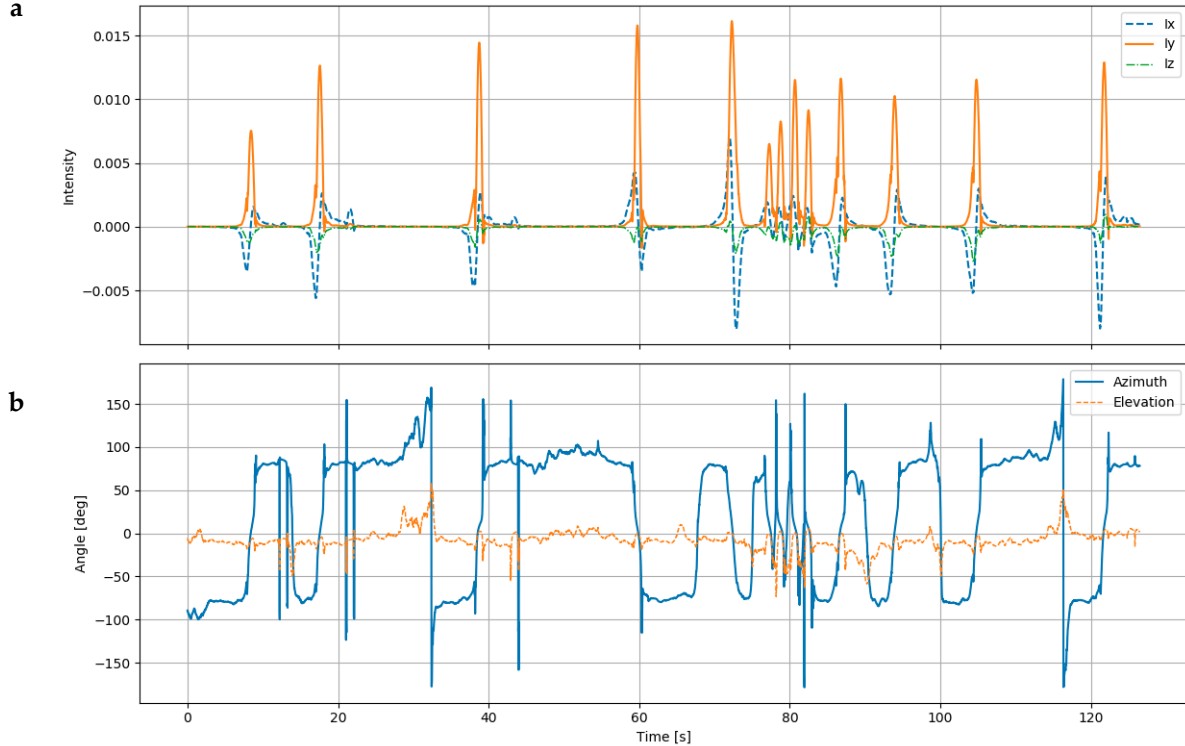

**Figure 13.** The results obtained from the AVS signal recorded for several vehicles: (**a**) intensity components, (**b**) azimuth and elevation.

The algorithm works in two stages. The first stage is based on the analysis of sound intensity signals and it detects acoustic events. The second stage analyses a detection function, based on the normalized source position, its task is to determine whether the acoustic event represents a vehicle

passing the sensor and detecting its direction of movement. The detected and verified acoustic events may be analyzed further, e.g., for the velocity estimation.

In the first stage, the intensity signal is analyzed with a sliding window **w** of length $N$, moving with a step equal to one sample. In the experiments, the window span was about 640 ms. It was found that using the intensity signal that was measured on the axis perpendicular to the road ($I_Y$) provides better results than the total intensity. An acoustic event is detected if:

$$\mathbf{w}_{N/2} = \max_i \mathbf{w}_i \text{ and } \frac{1}{N}\sum_{i=1}^{N} I_{Y(i)} \geq T_{\text{int}} \tag{13}$$

where $T_{int}$ is the minimum average intensity within the window and $I$ is the sample index within the window. The value of $T_{int}$ must be set according to the amplitude of the analyzed signal, so that both low-energy and short-term (impulse) acoustic events are discarded.

The second stage of the algorithm analyzes the changes in the normalized position of the sound source. Position $x$ of the source moving along the trajectory parallel to the X-axis of the system, at a normalized distance of $y = 1$ m, is equal to:

$$x = \tan(\phi) = \frac{I_Y}{I_X} \tag{14}$$

where $\phi$ is the azimuth of the source.

If a vehicle moves along this trajectory with an approximately constant velocity, then smooth changes in $x$ will be observed, and the direction of these changes will indicate the direction of the object's movement. For acoustic events that are not related to moving vehicles, much larger changes in the source position can be expected. Therefore, the detection metric $d$, as computed within the same window as in the first stage, is:

$$d = \frac{2}{N}\sum_{i=1}^{N/2}\left(\frac{I_{Y(i+1)}}{I_{X(i+1)}} - \frac{I_{Y(i)}}{I_{X(i)}}\right) \tag{15}$$

Only the first half of the window is used. In the case of isolated vehicles, the second half is redundant and, if several vehicles move close to each other, their measured intensities overlap, which usually causes more distortion in the second part of the window. The sign of $d$ indicates the direction of movement: vehicles moving towards positive $x$ values have $d < 0$, vehicles moving in the opposite direction have $d > 0$. Additionally, standard deviation within the window might be computed, similarly to $d$. It is expected that the standard deviation will be small for vehicles moving past the sensor, and high for unrelated acoustic events, which might be discarded with a maximum standard deviation threshold.

Figure 14 presents an example of detection. The maxima of the intensity function is detected as acoustic events (Figure 14a). The detection function (normalized $x$ position) smoothly changes within the acoustic events and oscillates randomly when no events are present (Figure 14b). The value of $d$ computed for each event indicates the direction of a vehicle's movement; this is marked with bars pointing upwards or downwards for $d < 0$ and $d > 0$, respectively.

Detections from the reference data are marked with dots, with the direction being indicated in the same way. It can be observed that most of the vehicles were correctly detected and identified. For isolated vehicles (frame 28429), the detection function changes smoothly within the whole detection window, and the analysis is straightforward. When multiple vehicles are moving close to each other, their detection functions overlap. In some events (e.g., frames 31804 & 31898), the results are correct. In the case of a heavy occlusion (multiple vehicles on both lanes), the probability of errors increases. In the presented example, the vehicle in frame 29835 is detected, but its direction is incorrect, due to the overlap of detection functions from many vehicles, and one vehicle (frame 29742) was missed, as two intensity maxima from two vehicles merged into a single one.

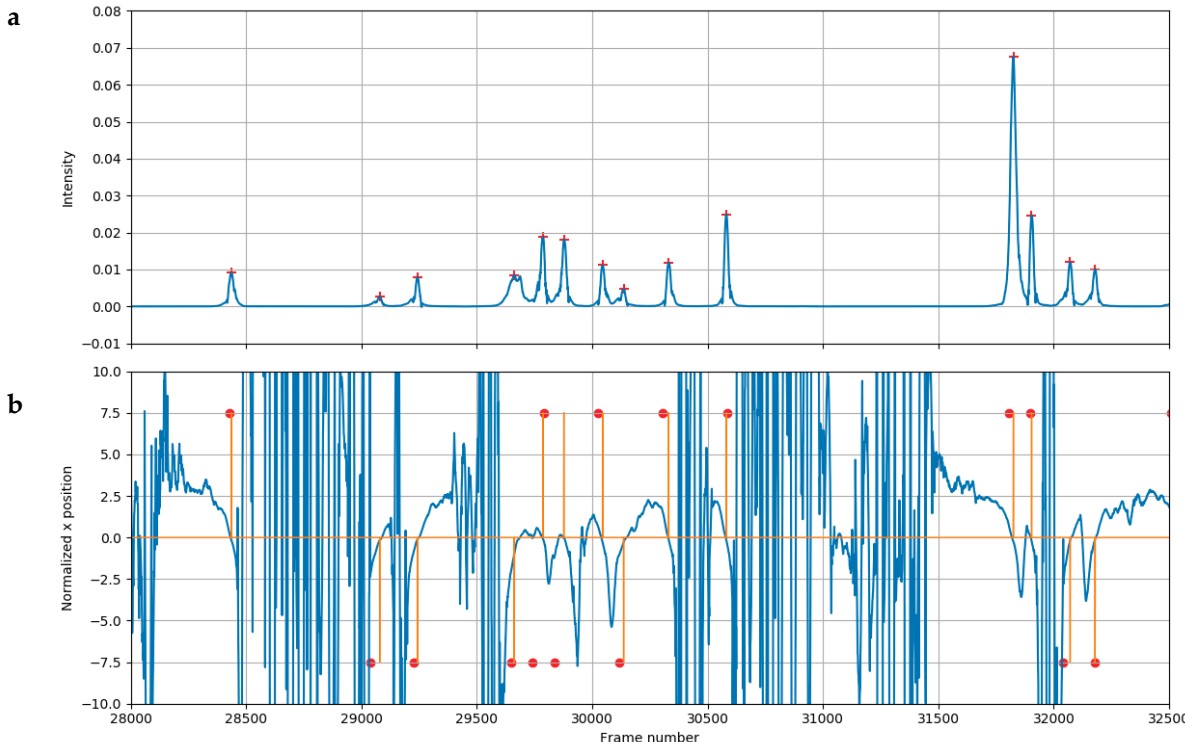

**Figure 14.** Example of the detection results: (**a**) sound intensity (line) and detected acoustic events (+); (**b**) detection function (line), detected vehicles (vertical bars, the direction of bars indicates the direction of movement), reference data (dots, vertical position indicates the direction of movement).

## 3. Results

### 3.1. Test Setup

A test setup was constructed and experiments were performed in a real-world scenario in order to verify the proposed algorithms. A low-cost RSM2650 Doppler sensor by B+B Sensors [25] was used. The sensor was connected through a custom-built amplifier and an analog-to-digital converter (48 kHz sampling rate) to a Raspberry Pi 3B microcomputer. All of the elements, together with an LTE router and a power supply, were placed in an enclosure (Figure 15). In the first stage of the research, signals from the sensor were recorded on the microcomputer and then downloaded for offline analysis. The analysis of the Doppler sensor signal was performed online on the microcomputer in the experiments described here, and the results (timestamp, velocity, and standard deviation) were available via the MQTT network protocol. The signal was analyzed in windows of 2048 samples (42.67 ms) with 75% overlap while using the Blackman window before FFT was computed. The processing algorithm was implemented in the Python programming language.

The AVS was constructed from six omnidirectional digital MEMS microphones, IvenSense INMP441 [26], operating at 48 kHz sampling rate with 24-bit resolution. Each microphone was mounted on a board of ca. $10 \times 10$ mm size that was connected through an I2S USB digital interface to a USB port on a computer (Figure 8). The sensor was mounted in a windshield at the bottom side of the enclosure. The six-channel signal was recorded on the microcomputer and then stored on a hard drive. The recordings were analyzed offline. Additionally, environmental sensors (temperature, pressure, precipitation, air quality), as well as a LiDAR sensor and a video camera, were mounted on the enclosure; these were not used in the experiments described here.

The test system was mounted on the outskirts of Gdańsk, Poland (near Leźno village), geographic coordinates: 54.344555, 18.443811. The monitored road section had one lane in each direction and the speed limit was 90 km/h. The measurements were performed on a straight and flat section of the road (Figure 16), where the typical speed of vehicles is 60 to 80 km/h. The enclosure was mounted 4 m away

from the road edge and the bottom side of the box was 2.9 m above the ground. The Doppler sensor was positioned at 3.2 m above the ground, oriented at 45° azimuth, and −18° elevation relative to the road axis. The algorithm only analyzed the closer lane (eastbound traffic).

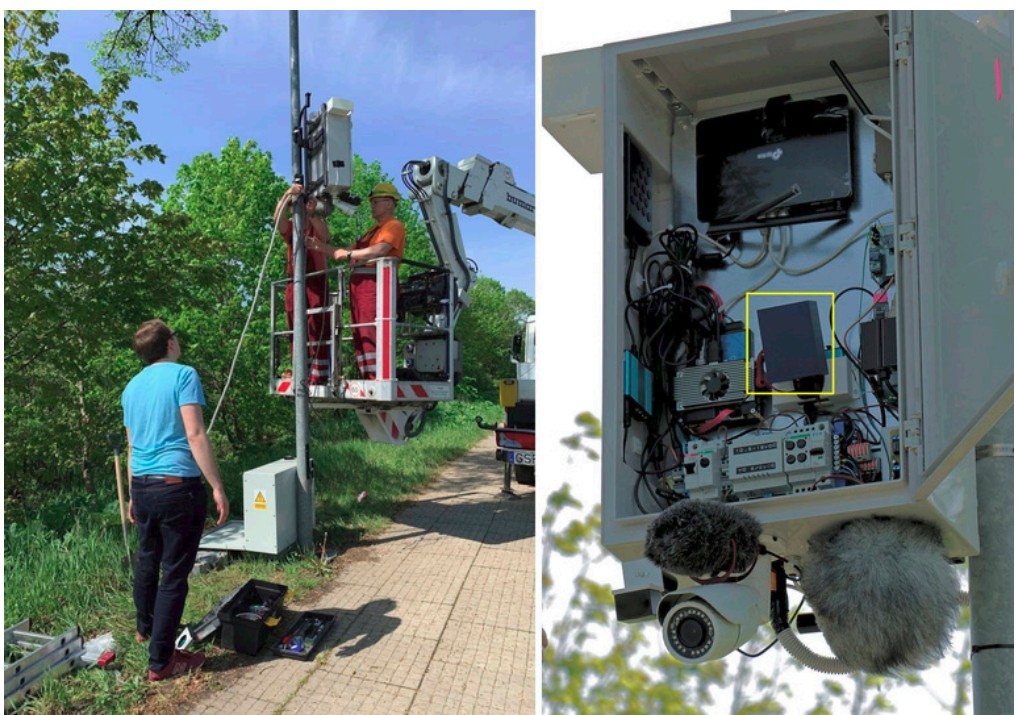

**Figure 15.** The test system mounted on the site. The Doppler sensor is located inside the box marked with the rectangle, the AVS in a windshield is visible at the bottom right, below the enclosure.

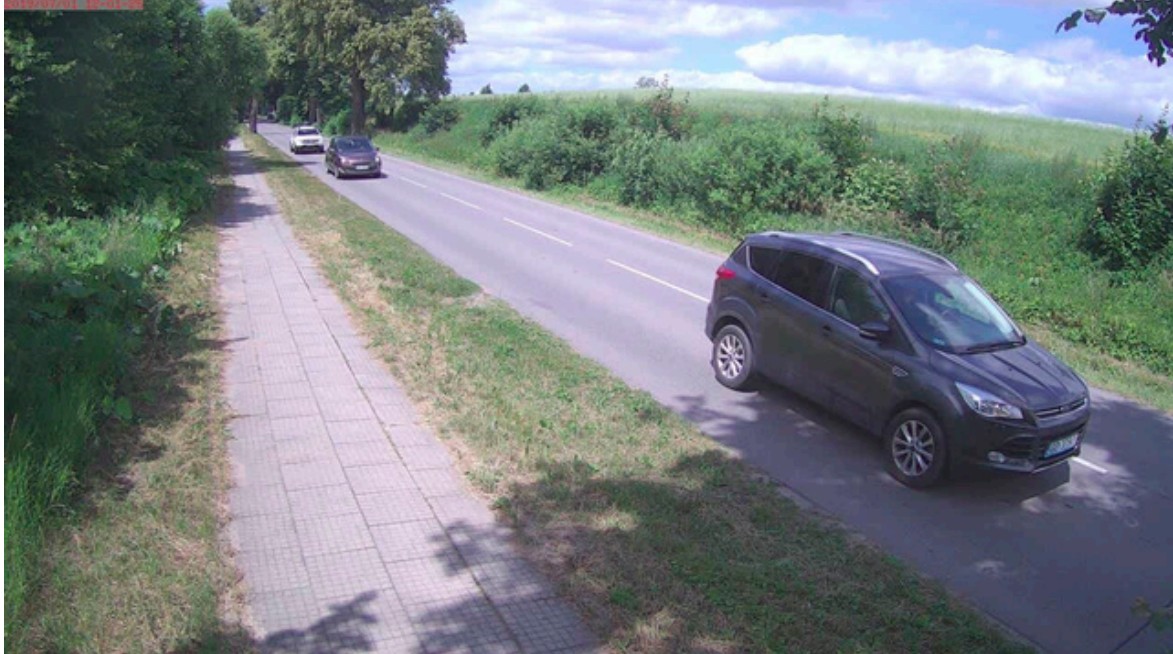

**Figure 16.** The test road section—a view from the camera mounted in the test system. The first measurement tubes are mounted at the trees visible in the back, the second pair is positioned near the bottom right corner of the photo.

A system based on pneumatic tubes (Metrocount MC5600 Vehicle Counter System) was mounted on the road in order to obtain reference data for the experiments. Two pairs of tubes were used. One pair was positioned near the test system, for comparison with the AVS results. The other pair of tubes were mounted ca. 100 m away, within the zone of Doppler sensor detection. Recordings that spanned a continuous period of 24 h (July 1st, 14:00 to 2nd July 2019) were obtained, and timestamps and velocity measured for each vehicle (from both lanes) were used for comparison with the Doppler and AVS sensors results. The temperature during the recording was 15 °C to 26 °C, average pressure was 997 hPa, the wind was up to 7 m/s from the West, and there were occasional periods of rainfall (about 15% of the total time).

### 3.2. Analysis of Vehicle Counting

Aggregating the detection results in 30-min. periods was undertaken to analyze the results of vehicle counting on the lane closer to the test setup. Data from the tubes were used as the reference and it is assumed that there are no detection errors in the recorded data (this was partially confirmed by reviewing selected sections of the recorded video). Table 1 presents the results that were obtained for both sensors within the measurement period. For the AVS, a total of 30 min. was missing from the recorded material due to technical difficulties. Figure 17 shows the vehicle count aggregated in 30 min. intervals, for the data from the Doppler sensor, analyzed by the proposed algorithm, and data from the reference device. Figure 18 presents a similar plot calculated for the AVS. The Pearson correlation coefficient between the sets of the calculated and the reference vehicle counts is 0.994 for the radar and 0.995 for the AVS.

**Table 1.** Summary of the vehicle detection results.

| Sensor | Doppler | AVS |
|---|---|---|
| Analyzed time | 24 h | 23 h 30 min |
| Total number of vehicles | 2998 | 2953 |
| True detections | 2742 | 2583 |
| False negatives | 256 | 370 |
| False positives | 44 | 189 |
| Recall | 91.46% | 87.47% |
| Precision | 98.42% | 93.18% |
| Accuracy | 90.14% | 82.21% |

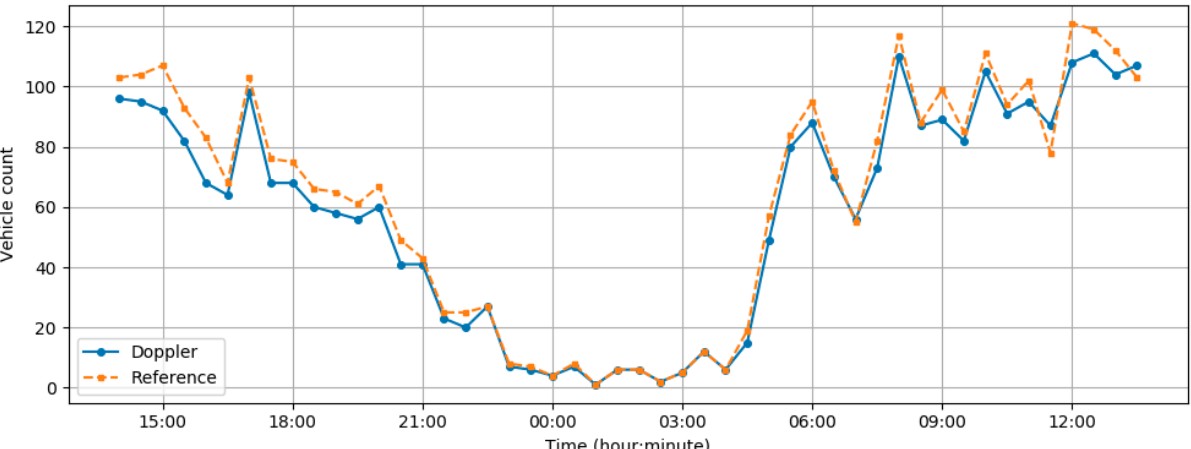

**Figure 17.** Vehicle count in 30-min. intervals—measured with the Doppler sensor and the proposed algorithm (solid line), and the reference data from the tube detector (dashed line).

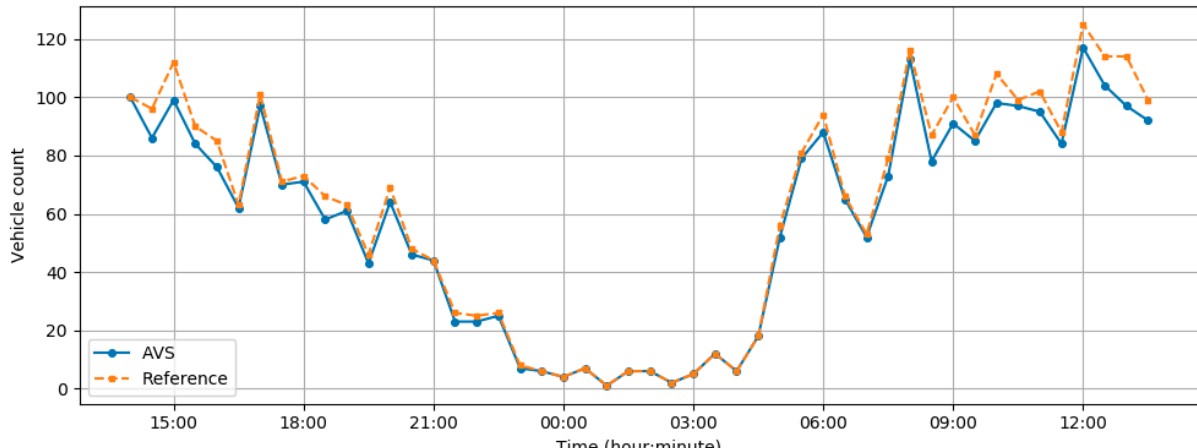

**Figure 18.** Vehicle count in 30-min. intervals—measured with the AVS and the proposed algorithm (solid line), and the reference data from the tube detector (dashed line).

'Ghost tracks' when the algorithm followed noisy components of the signal and produced duplicated detections mainly caused the observed false-positive results. To conclude, improving the algorithm accuracy requires its modifications that will make it more robust to the observed errors, and also tuning the algorithm parameters and, if possible, repositioning the sensor.

The results of vehicle counting that were obtained with the AVS and the proposed algorithm (Figure 18) were consistent with the results from the Doppler sensor. The number of false-positive and false-negative results is slightly larger for the AVS, the overall accuracy is lower (82% vs. 90%). The main source of incorrect detection results is the problem with determining the direction of movement in the case of occlusion (vehicles on both lanes, moving in opposite directions) and when several vehicles move close to each other. In such cases, the detection function does not allow for accurate direction analysis, because the signals from multiple vehicles overlap. As a result, some vehicles were detected, but their direction was incorrect, as shown in Table 2. However, this problem occurs on both lanes. In the presented experiment, the number of vehicles on each lane was similar, so that false positive and false negative results on each lane were mostly balanced. Some vehicles could not be detected at all, which was mostly due to high occlusion. As shown in Table 2, the number of vehicles in the closer lane that was not detected is almost equal to the number of vehicles detected with incorrect direction. The number of vehicles that were not detected is similar on both lanes, while the incorrect detection of the direction happens more often on the closer lane. The number of false detections is higher in the further lane, which results from the occlusion. In total, statistics that were obtained in 30-min. intervals were very similar to those from the Doppler sensor, they also slightly underestimate the real vehicle count in the case of high traffic. The trend also highly correlates with the reference data.

**Table 2.** Detailed analysis of vehicle detection in the AVS signal, on both lanes.

| Lane | Closer Lane | Further Lane | Both Lanes |
|---|---|---|---|
| Number of vehicles | 2953 | 2940 | 5893 |
| Detected, correct lane | 2583 | 2691 | 5274 |
| Detected, wrong lane | 191 | 80 | 271 |
| Not detected | 179 | 169 | 348 |
| False detections | 109 | 190 | 299 |

*3.3. Analysis of Velocity Measurement Using Doppler Sensor*

Comparison of velocity measured by the proposed algorithm analyzing signals from the Doppler sensor and by the reference device cannot be accurately performed due to the fact that the tube-based system measured the velocity at one point, about 100 m from the sensor, while the Doppler sensor

measured velocity at different points within the zone approximately 50 to 100 m from the sensor. Therefore, any observed differences may be caused both by measurement errors and by vehicles that accelerate or brake within the detection zone. The mean squared difference (MSD) between the measurements of individual vehicles by both sensors is 19.45, with a standard deviation that is equal to 176.47, and the root of the MSD (RSMD) is 4.41 ± 13.28 km/h. The accuracy of measuring the velocity of individual vehicles with the proposed algorithm is satisfactory while taking the condition mentioned before into account.

Figure 19 shows the results of averaging the velocity in 30-min. intervals for both data sources. It can be observed that the results that were obtained from the evaluated algorithm are slightly lower than for the reference device (MSD 1.93 ± 1.68, RMSD 1.39 km/h). Both datasets follow a similar trend and the Pearson correlation coefficient is 0.92. This confirms that the evaluated algorithm provides velocity measurements with accuracy that is sufficient for collecting traffic statistics.

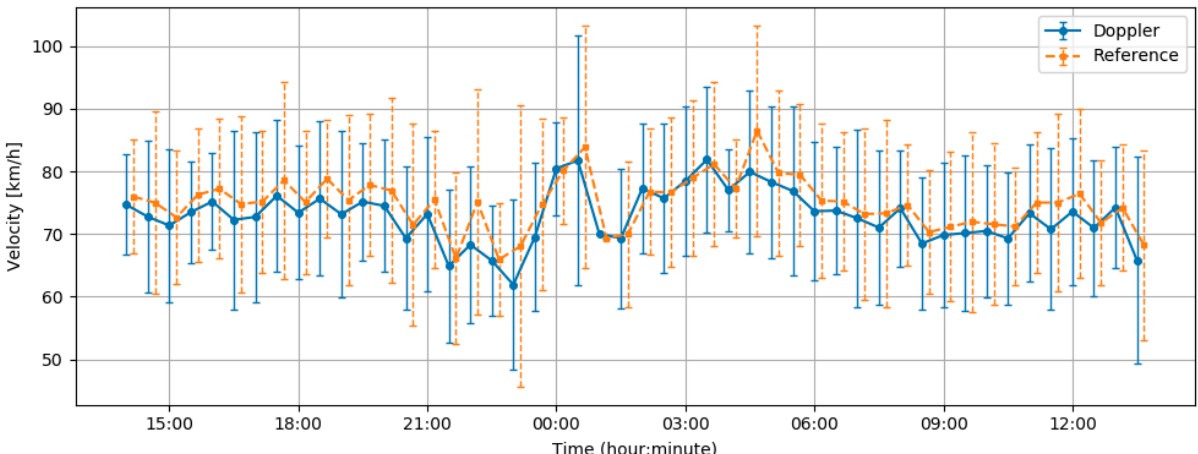

**Figure 19.** Vehicle velocity in 30-min. intervals—measured with the Doppler sensor and the proposed algorithm (solid line) and reference data from the tube detector (dashed line, shifted on the time axis for improved readability). Points represent mean velocity, error bars—standard deviation.

### 3.4. Analysis of Velocity Measurement Using Acoustic Vector Sensor

The vehicle velocity measurements using the Acoustic Vector Sensor are not so obvious as for the Doppler sensor. In the beginning, we need to mention that the AVS gives as the possibility to localize and tracking of moving sound sources. Let us consider a single point sound source moving through a linear trajectory, parallel to the X-axis of the sensor, with a constant velocity. The position of the source is $P(t) = (x(t), y)$, where $y$ is constant. The velocity of the source might be expressed as:

$$v = \frac{\Delta x}{\Delta t} = \frac{\Delta(y \cdot \tan \phi)}{\Delta t} = \frac{y}{\Delta t}\Delta\left(\frac{I_X}{I_Y}\right) \tag{16}$$

where $\phi$ is the azimuth measured by the sensor.

Real vehicles are not single sound sources, but rather a setup of several sources (tires, engine, exhaust, etc.). Distances between sources, differences in the source power, and in the directivity of each source all contribute to the obtained results because the vehicles move close to the sensor. It was confirmed by a computer simulation, in which the azimuth that was computed for a single source and a setup of four sources, moving with the same velocity, was calculated. The obtained results (Figure 20), which are consistent with the measurements from the real AVS, indicate that the azimuth that was obtained for multiple sources moving together (solid line) deviates from the tangential line of the single-source case (dotted line). This is caused by an additional velocity component, representing the movement of a virtual sound source within a vehicle, which is nonlinear, with the opposite direction to the velocity vector of the moving vehicle. The AVS measures the position of a virtual sound source,

so both velocity components are included. As a consequence, the velocity estimate computed by the algorithm (which assumes a single source) is lower than the real velocity. The difference depends on a number of factors, such as the width and the length of a vehicle, distance from the sensor, relative power of each source, and directivity of the sources. These variables are unknown, so it is not possible to correct the obtained velocity estimates.

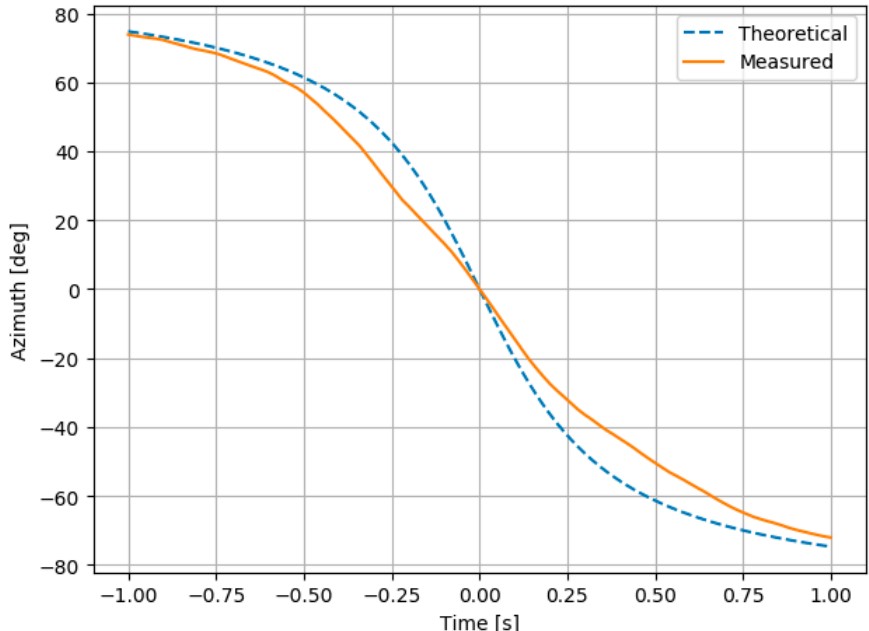

**Figure 20.** Computer simulation of the azimuth for a single source (dotted line) and an azimuth measured by the sensor (solid line).

## 4. Discussion

The accuracy of the proposed algorithm for vehicle counting in the Doppler sensor signals is approximately 90%, which is low for systems that are intended to detect each vehicle (e.g., traffic law enforcement systems), but it is sufficient for the intended application of collecting traffic statistics. Two main reasons for false-negative results were identified during the analysis of the obtained results. The first problem is common for all radar-based detection systems: it is difficult to distinguish each vehicle driving in a 'train', where several vehicles move close to each other. The most common error in the obtained result was skipping one vehicle in a sequence of four or more vehicles moving almost 'bumper to bumper'. The problem is that since all of the vehicles move with similar velocity, tracks of these vehicles overlap, and, in some cases, the tracking algorithm merges one track with another. It can be observed in Figure 17 that false-negative results mainly occur during the rush hours, which confirms these conclusions. One possible solution to this problem, left for future research, is implementing an additional algorithm that analyzes low-frequency components (final sections of the tracks) and detects vehicles moving past the sensor. Such an algorithm is not trivial, as noise and slow objects distort low frequencies (bicycles, pedestrians). The second problem was related to cars moving with high velocity, above 100 km/h. In the test system, the sensor was positioned too far from the road (due to practical constraints), which resulted in a low signal-to-noise ratio, causing a loss of the initial section of the tracks. For vehicles moving with high velocity, some tracks were too short to provide valid measurements. Moving the sensor closer to the road, and by decreasing the minimal track length, which in turn may increase the number of false-positive results, can mitigate this problem.

As discussed earlier, it is not possible to measure the exact velocity of individual cars with the AVS, as the obtained values underestimate the real velocity. Nevertheless, an attempt was made to calculate the averaged velocity within the time slots. For this purpose, the velocity of each vehicle was estimated within sections of the AVS signal, as determined by the vehicle detector. The distance between the

sensor and the vehicles was chosen as 5.5 m for the closer lane and 8.5 m for the further one—a midpoint of each lane was selected as an average distance to the sound source. In some cases, due to occlusion by multiple vehicles present in the detection zone at the same time, the velocity estimation was not possible in some cases. In fact, as much as 11% of the detected vehicles were discarded from the analysis. The remaining speed estimates were averaged within 30-min. periods. The results are shown in Figure 21. Just like the results obtained for the individual vehicles, the averaged values are significantly lower than the real velocity from the reference device. However, the obtained pseudo-velocity function correlates with the reference data and Pearson's coefficient is 0.78. Therefore, the obtained values may be used as an indicator of relative velocity changes, allowing e.g., for detection of periods in which the averaged velocity falls below a selected limit, even if the actual velocity cannot be measured with the AVS. Moreover, the ratio of the obtained estimates to the reference data, as computed for each time slot, is 0.78 ± 0.03 (mean ± standard deviation), so it is stable within the whole observation period. A more accurate estimate of the real velocity is obtained if the obtained estimates are multiplied by the scaling factor equal to the mean ratio. The results of the experiments indicate that this scaling factor mainly depends on the distance between the sensor and the road, as other factors, related to individual vehicles (width, height, and velocity of a vehicle, position on the lane) are averaged and they do not influence the result. Therefore, the scaling factor might be potentially determined by measuring the distance between the sensor and the road. This aspect requires further research.

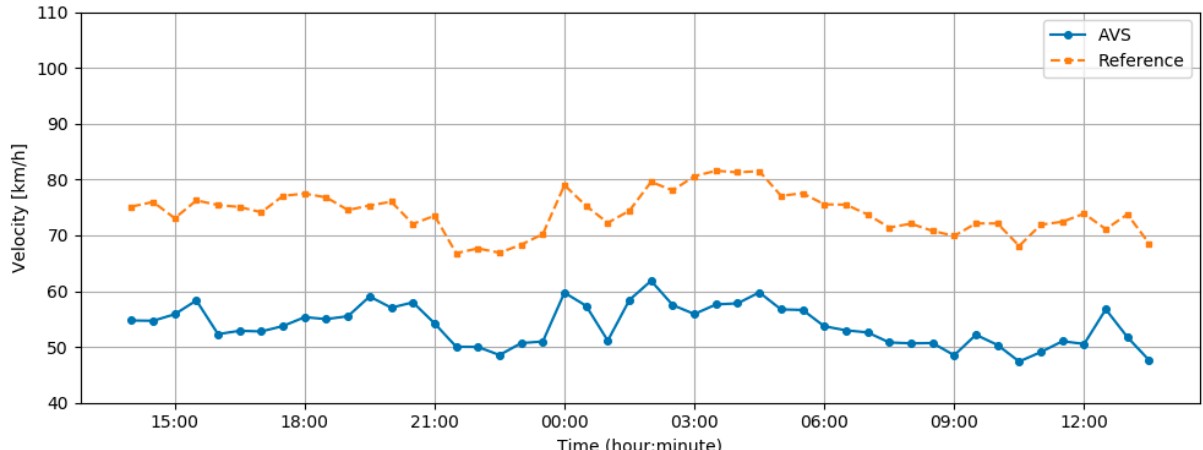

**Figure 21.** Velocity estimation from the AVS signal, averaged in 30-min. periods, as compared with the reference data.

The main advantage of the Doppler sensor is that it is capable of accurate measurement of the velocity of both individual vehicles and time-averaged statistics. The AVS is not able to achieve that, because the position of an apparent acoustic source within a vehicle is not constant. The correction of this effect is a separate problem that is outside of the scope of this paper and it is left for future research. Nevertheless, after applying an empirically found scaling coefficient, the AVS is able to provide traffic statistics with sufficient accuracy for the intended purpose, i.e., determining the trends in traffic speed variations and detection of periods in which the averaged velocity falls below a selected limit.

The vehicle counting accuracy relative to the traffic intensity is similar for both methods; they underestimate the vehicle count in high traffic scenarios. Both methods achieve accuracy close to 100% if the traffic intensity is below 100 vehicles per hour.

Although the acoustical vector sensor (AVS) has a lower accuracy than Doppler in vehicle counting and it is not able to measure the vehicle speed accurately, it has some advantages over the Doppler sensor. Namely, it does not emit any signals, it is not susceptible to electromagnetic interferences, and it allows for further analysis of audio signals, such as the assessment of the road surface state (e.g., wet/dry).

## 5. Conclusions

The authors developed and examined two methods for estimating traffic intensity in real traffic conditions. The first method is based on a microwave Doppler sensor. This method requires that the sensor emits a microwave signal, so it is an active method. This sensor is susceptible to electromagnetic interference from nearby sources, such as power lines, radar systems, cellular network base stations, etc. Additionally, waves that are emitted by the sensor may interfere with other devices operating in the same frequency range. The second method is based on an analysis of sound intensity vectors by means of the acoustic vector sensor. This method passively analyses acoustic signals that are emitted by moving vehicles. The sensor is not affected by electromagnetic interference, but acoustic noise might influence its operation. Both of the methods were evaluated in a real-world scenario while using a reference system based on pneumatic tubes. The conclusions are, as follows.

The vehicle counting accuracy of the proposed algorithm using the Doppler sensor signals is approximately 90%, which is sufficient for the intended application of collecting traffic statistics. The observed errors result from factors that are common for all radar-based measurement systems, namely the occlusion of multiple objects, inability to distinguish vehicles moving close to each other with similar speed, and duplicated detections for some vehicles. Modifications of the proposed algorithm and tuning its parameters to improve the detection accuracy in difficult situations, such as those observed in the experiments, is a topic of future research.

For the acoustic vector sensor, the proposed algorithm for vehicle counting provided results that are consistent with these from the Doppler sensor, with a slightly larger number of false-positive and false-negative results, and lower overall accuracy (82% vs. 90%). The main problem is related to determining the direction of movement in the case of occlusion (vehicles on both lanes, moving in opposite directions) and when several vehicles move close to each other. In such cases, an accurate direction analysis is problematic because the signals from multiple vehicles overlap. As a result, vehicles are detected, but their direction of movement is not correctly recognized. The algorithm also slightly underestimates the real vehicle count in case of high traffic. These problems will be addressed in future research.

Our observations that were collected in the course of experimental studies show that microwave sensors and acoustic sensors have application prospects for measuring traffic in order to discover traffic congestions by autonomous road signs or in other traffic measuring systems. The Doppler radar that we have improved and the constructed and the calibrated acoustic probe are applicable to perform vehicle detection and tracking, as well as vehicle speed measurement.

Both methods may be used for statistical analysis of traffic intensity and speed, providing valuable data for automated traffic management systems. The main advantage of the acoustic sensor over the microwave sensor is that it does not require sending any signals that could interfere with nearby devices, and it is not affected by any sources of electromagnetic signals in the vicinity.

The future work will focus on the optimizations of the proposed algorithms, which will lead to the increased accuracy of vehicle counting and velocity measurement. In the case of the AVS, the main topic of the research will be related to improving the algorithm for the detection of vehicle direction, and on developing a method for correction of the moving apparent sound source, which will allow for velocity measurement with the AVS.

**Author Contributions:** Conceptualization, A.C.; methodology, G.S., and J.K.; software, G.S. and J.K.; validation, G.S. and J.K.; formal analysis, A.C.; J.K.; G.S.; investigation, G.S. and J.K.; writing—original draft preparation, G.S. and J.K.; writing—review and editing A.C.; visualization, G.S.; supervision, A.C.; project administration, A.C.; funding acquisition, A.C. All authors have read and agreed to the published version of the manuscript.

**Funding:** Project financed by the Polish National Centre for Research and Development from under the EU Operational Programme Innovative Economy No. POIR.04.01.04-0089/16 entitled: INZNAK—"Intelligent road signs with V2X interface for adaptive traffic controlling".

**Conflicts of Interest:** The authors declare no conflict of interest.

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
