# Peer review of "Estimating Traffic Intensity Employing Passive Acoustic Radar and Enhanced Microwave Doppler Radar Sensor"

_remotesensing, doi:10.3390/rs12010110_

Round 1
Reviewer 1 Report
the paper entitled "estimating traffic intensity employing passive acoustic radar and enhanced microwave doppler rada sensor" proposes two methods for estimating traffic intensity. The methods are assessed in real conditions.
The research issue is clearly presented and the article is written in a clear and well illustrated manner. The paper is composed of five sections.
Section 1 introduces the subject and proposes a state-of-the-art with appropriate and very recent references. the authors have positioned their contribution in relation to these works.
Section 2 is dedicated to Materials and methods with two subsection (approaches using doppler or acoustic sensors). In section 2.2.1, the orientation of axis u(x, uy, uz) related to the vehicle or road is unclear. More explanations, such as a diagram, would be welcome.
line 222 : below a threshold -> which one ?
line 248 : Three parallel axes -> i do not understand, it is not orthogonal axes ?
In section 3 , the authors show the results. In figure 14, i am not sure to visualize the dark blue box. For a better visualization, the authors could perhaps insert a bounding box to highlight the sensor in the image.
Section 4 and 5 are dedicated to discussion and conclusion
Nevertheless, I have a general question: From the results, it seems that the Doppler sensor is generally better suited for this type of measurement. I understand that there are difficulties (active signal, interference), but is this really detrimental to this type of statistical analysis? Will even by improving the processing of acoustic data, the same level of results as the Doppler signal be achieved?
Minor corrections or suggestions:
line 66 : they proposed the proposed system -> they proposed a system (...) line 96 : systems , that ??? line 146 : it's positioning --> its positioning ? line 283 : intensity calculations -> maybe "intensity computations" ? line 316 (figure 10) : Fraquency -> Freqency line 323 : development the vehicle -> development of the vehicle lline 324 : 120 second -> 120 seconds
Author Response
The authors wish to thank the Reviewer for a thorough analysis of the manuscript and for helpful remarks. The indicated errors were corrected, and the suggested improvements were added to the text. Answers to specific questions are given below.
Section 2 is dedicated to Materials and methods with two subsections (approaches using Doppler or acoustic sensors). In section 2.2.1, the orientation of the axis (ux, uy, uz) related to the vehicle or road is unclear. More explanations, such as a diagram, would be welcome.
Figure 10 was added.
line 222: below a threshold -> which one?
A comment was added to the text. The threshold used in the experiments is 20 km/h (c.a. 500 Hz). If the track goes down below this threshold, it means that the vehicle moved close to the sensor. If the track ends at a higher frequency, then it does not represent a vehicle that drove near the sensor.
line 248: Three parallel axes -> i do not understand, it is not orthogonal axes ?
That was an obvious mistake, the axes are orthogonal. The error was corrected.
In section 3, the authors show the results. In figure 14, I am not sure to visualize the dark blue box. For better visualization, the authors could perhaps insert a bounding box to highlight the sensor in the image.
We added a rectangle showing the sensor position.
Nevertheless, I have a general question: From the results, it seems that the Doppler sensor is generally better suited for this type of measurement. I understand that there are difficulties (active signal, interference), but is this really detrimental to this type of statistical analysis? Will even by improving the processing of acoustic data, the same level of results as the Doppler signal be achieved?
Our aim is to develop a distributed network of sensors that collect traffic statistics at numerous points at once. Therefore, the monitoring stations need to be able to count vehicles and to measure their speed at any location that is necessary for the traffic analysis systems. In our experiments, we observed that using Doppler sensors is practically impossible at some locations. e.g. near an airport, a power distribution station or below high voltage power lines. Doppler sensors are susceptible to electromagnetic interference, to the point that the distortions mask the useful signals and make the detection and measurements impossible. Therefore, we aim to use an alternative solution, namely the acoustic vector sensor, at locations in which Doppler sensors cannot be used. The aim of the paper was to answer the question: “may the AVS be useful for the described scenario?”. The presented results of experiments indicate that although the AVS is not as accurate as of the Doppler sensor, it may be sufficient for collecting traffic data, and it is not susceptible to electromagnetic interference.
Minor corrections or suggestions:
All errors were corrected.
Reviewer 2 Report
This manuscript presents some experiments using an active K band Doppler sensor, and a passive acoustic sensor to measure traffic intensity.
I am quite unsure what to recommend with regard to its suitability for publication. There is very little scientific contribution, and the contribution that is made is rather empirical, rather than theoretically based. But on the other hand, there is clearly a lot of work involved in setting up experiments like this, and they seem to have been done thoroughly, and the results could well be of interest to some readers.
Perhaps my strongest objection is that the methods were not well justified. In particular, the pneumatic tubes were described as 'cumbersome and only suitable for permanent installations.' I cannot see why this is the case, and given their accuracy and reliability, I would have thought it would be easier to spend a tiny bit of effort into making the pneumatic tubes less cumbersome and more suitable for temporary deployment, rather than devising new methods which are less reliable.
Regarding Figure 5; since you have I and Q channels, you should be able to separate negative frequencies directly, and create a spectrogram that includes negative frequencies. Wouldn't this help with discriminating between vehicles traveling in opposite directions?
Regarding Figure 8; a) Why doesn't diffraction around the circuit boards interfere with the velocity measurement? b) Why doesn't reflection from the inside of the windscreen interfere with the results?
Would making the acoustic vector sensor larger help with its performance? Or perhaps have two sensors separated in the direction of traffic flow? Would that help to distinguish between the two directions?
The English is mostly comprehensible, but there are many very awkward sentences.
Some minor comments:
l26 V2X in the introduction should be explained.
l95 is incomplete
Lines 193-197 would flow better if they were near the start of Section 2.1.2.
l248 Surely the word 'parallel' should be 'orthogonal' ?
l339 'better results' How much better? Can you quantify that?
Figure 11: an additional figure to show how x,y, and z are oriented would be helpful for understanding the description.
Author Response
The authors wish to thank the Reviewer for a thorough analysis of the manuscript and for helpful remarks. The indicated errors were corrected, and the suggested improvements were added to the text. Answers to specific questions are given below.
Perhaps my strongest objection is that the methods were not well justified. In particular, the pneumatic tubes were described as 'cumbersome and only suitable for permanent installations.' I cannot see why this is the case, and given their accuracy and reliability, I would have thought it would be easier to spend a tiny bit of effort into making the pneumatic tubes less cumbersome and more suitable for temporary deployment, rather than devising new methods which are less reliable.
Pneumatic tubes are only suitable for temporary measurements, as they obstruct the traffic, so they are only installed for several days. The aim of our work is to construct a network of monitoring stations that perform continuous traffic analysis. Inductive loops are used for constant monitoring, but they are mostly used at urban intersections. Moreover, they are expensive, and they require invasive installation on the road surface. As we are interested in traffic statistics and not in measuring individual vehicles (so we don’t need very high reliability at a high cost), we believe that sensors such as the Doppler sensor and the AVS are suitable for the task.
Regarding Figure 5; since you have I and Q channels, you should be able to separate negative frequencies directly, and create a spectrogram that includes negative frequencies. Wouldn't this help with discriminating between vehicles traveling in opposite directions?
Both channels, I and Q, contain signals reflected from the vehicles moving in both directions. Their amplitude spectra are almost identical, the channels only differ in phase spectra. Therefore, separation of the traffic directions requires picking spectral components based on their interchannel phase. Thanks to that, the problem of occlusion, when vehicles moving in opposite directions overlap, is significantly reduced.
Regarding Figure 8; a) Why doesn't diffraction around the circuit boards interfere with the velocity measurement? b) Why doesn't a reflection from the inside of the windscreen interfere with the results?
In our previous research, we have found that the windscreen used in our installation, described in the paper, does not disturb the calculation of DOA values. The reason is that the instantaneous intensity from the vehicles passing by the sensor has a noise-like character. It means that the interrelation of the acoustic signal around the sensor and windscreen can be neglected. Moreover, we limited the frequency of the analyzed sound intensity to 4 kHz. It means that the wavelength of the highest frequency is equal to 340 / 4000 = 0.085 [m]. The sensor is placed inside the windscreen, in its middle point. It means that the windscreen dimensions are small in comparison with the wavelength, even for the highest analyzed frequency. For these reasons, the windscreen does not disturb the measurements.
Would making the acoustic vector sensor larger help with its performance? Or perhaps have two sensors separated in the direction of traffic flow? Would that help to distinguish between the two directions?
Thank you for these suggestions. First of all, we have assumed that the traffic flow will be analyzed using a single AVS. We applied our own AVS based on the p-p probe, for determination of the particle velocity. In our design, the distance between the microphones in each pair is 10 mm. This distance is sufficient for the proper determination of the intensity for frequencies up to 6 kHz. The larger distance between the microphones would enlarge the whole sensor (we try to avoid this disadvantage) and it has an essential influence on the high-frequency limit. For example, for frequencies up to 4kHz, the distance between the microphones in the intensity probe should not exceed 20 mm.
In our future work, we will consider the application of more than one AVS to improve the functionality of the device, especially for improved detection of the direction of moving vehicles, and their speed.
Some minor comments:
The errors were corrected in the manuscript.
l339 'better results' How much better? Can you quantify that?
We cannot quantify that, we just have observed in the intensity plots that if we used the total intensity, the detection was more problematic in terms of finding the correct thresholds, as the intensity from the parallel and the vertical axes influenced (“smeared”) the detection function. Once we decided to use only the perpendicular axis intensity, the detection function marked the vehicle presence more prominently, and the analysis became easier. therefore, we concluded that the intensity determination from the other two axes was not needed for the detection of vehicle presence.
Figure 11: an additional figure to show how x,y, and z are oriented would be helpful for understanding the description.
We added a new Figure 10.
Reviewer 3 Report
The presented paper describes a novel approach to sense traffic by means of embedded acoustic arrays and dopler sensing. The results are very promissing and we encourage the authors to further improve the described method. Especially the approach for acoustic imaging and sensing is notable.
Congrats for the nice work
Author Response
Thank you for reviewing our manuscript.